# Step-by-step causal analysis of EHRs to ground decision-making

**Matthieu Doutreligne** [1,2]*, **Tristan Struja**[3,4], **Judith Abecassis**[1], **Claire Morgand**[5], **Leo Anthony Celi**[3,6,7], **Gaël Varoquaux**[1]

**1** Soda Team, Inria Saclay, Palaiseau, France, **2** Mission Data, Haute Autorité de Santé, Saint-Denis, France, **3** Laboratory for Computational Physiology, Massachusetts Institute of Technology, Cambridge, Massachusetts, United States of America, **4** Medical University Clinic, Division of Endocrinology, Diabetes & Metabolism, Kantonsspital Aarau, Aarau, Switzerland, **5** Agence Régionale de Santé Ile-de-France, Saint-Denis, France, **6** Division of Pulmonary, Critical Care and Sleep Medicine, Beth Israel Deaconess Medical Center, Boston, Massachusetts, United States of America, **7** Department of Biostatistics, Harvard T.H. Chan School of Public Health, Boston, Massachusetts, United States of America

* m.doutreligne@has-sante.fr

## Abstract

Causal inference enables machine learning methods to estimate treatment effects of medical interventions from electronic health records (EHRs). The prevalence of such observational data and the difficulty for randomized controlled trials (RCT) to cover all population/treatment relationships make these methods increasingly attractive for studying causal effects. However, researchers should be wary of many pitfalls. We propose and illustrate a framework for causal inference estimating the effect of albumin on mortality in sepsis using an Intensive Care database (MIMIC-IV) and comparing various sensitivity analyses to results from RCTs as gold-standard. The first step is study design, using the target trial concept and the PICOT framework: Population (patients with sepsis), Intervention (combination of crystalloids and albumin for fluid resuscitation), Control (crystalloids only), Outcome (28-day mortality), Time (intervention start within 24h of admission). We show that too large treatment-initiation times induce immortal time bias. The second step is selection of the confounding variables based on expert knowledge. Increasingly adding confounders enables to recover the RCT results from observational data. As the third step, we assess the influence of multiple models with varying assumptions, showing that a doubly robust estimator (AIPW) with random forests proved to be the most reliable estimator. Results show that these steps are all important for valid causal estimates. A valid causal model can then be used to individualize decision making: subgroup analyses showed that treatment efficacy of albumin was better for patients >60 years old, males, and patients with septic shock. Without causal thinking, machine learning is not enough for optimal clinical decision on an individual patient level. Our step-by-step analytic framework helps avoiding many pitfalls of applying machine learning to EHR data, building models that avoid shortcuts and extract the best decision-making evidence.

**Data availability statement:** The datasets are available on PhysioNet (https://doi.org/10.13026/6mm1-ek67). We used MIMIC-IV.v2.2 The code for data

preprocessing and analyses are available on github https://github.com/soda-inria/causal_ehr_mimic/.

**Funding:** TS received fundings from (Swiss National 293 Science Foundation, P400PM_194497 / 1). GV has been granted the ERC 294 (Intercept-T2D HORIZON-HLTH-2022-STAYHLTH-02-01). LAC is funded by the 295 National Institute of Health through NIBIB R01 EB017205. The funders had no role in study design, data collection and analysis, decision to publish, or preparation of the manuscript.

**Competing interests:** Leo Anthony Celi is part of the editor board of *PLOS Digital Health*.

## Author summary

Rich routine-care data, as EHR or claims, is useful to individualize decision making using machine learning; but guiding interventions requires causal inference. Unlike with an RCT, interventions in routine data do not easily enable an apple-to-apple measure of the effect of an intervention, leading to many analytical pitfalls, particularly in time-varying data. We study these in a tutorial spirit, making the code and data openly available. We give 5 analytical steps for data-driven individualized interventions: Step 1) Study design, where common pitfalls are selection bias, with information unequally collected across treatment and control patients, and immortal time bias, where the inclusion-defining event interacts with the intervention time. Step 2) Identification of the causal assumptions and categorization of confounders. Step 3) Estimation of the causal effect of interest by correct aggregation of confounders and selection of an appropriate statistical model. Step 4) Assessing the analysis' robustness to assumptions, and finally Step 5) Individualizing treatment decision, by exploring treatment heterogeneity, eg across subgroups. Studying choice of fluid resuscitation in sepsis, we show that common mistakes in steps 1, 2, and 3 equally compromise causal validity.

## Introduction: Data-driven decisions require causal inference

Informing a care option extends beyond merely predicting the occurrence of an event; it involves estimating the effect of the corresponding treatment effects. Routine-care data comes naturally to mind to guide routine decisions, but they require care to estimate treatment effects as they are observational, unlike Randomized controlled trials (RCTs). This context calls for causal inference statistical frameworks. But merely applying these tools to the data does suffice to ensure the validity of the inferences; numerous considerations must be carefully addressed.

### Individualized medicine and machine learning challenges

Machine learning plays a pivotal role in individualized medicine [1–5]. It demonstrated superior performance over traditional rule-based clinical scores in predicting a patient's readmission risk, mortality, or future comorbidities using Electronic Health Records (EHRs) [1–5]. However, mounting evidence suggests that machine-learning models can inadvertently perpetuate and exacerbate biases present in the data [6], including gender or racial biases [7,8], and the marginalization of under-served populations [9]. These biases are typically encoded by capturing shortcuts—stereotypical or distorted features in the data [10–12]. For instance, numerous machine learning algorithms rely on post-treatment information [13–16], exemplified by a diagnostic model for skin cancer that depends on surgical marks [11]. For Intensive Care Unit data, focus of our study, such information markedly improves mortality prediction (S1 Fig), but cannot inform decisions.

### The importance of causal reasoning in data-driven decision-making [17]

While conventional machine learning relies on retrospective to generate predictions of future effects [18], truly informing decision-making needs a comparison of potential outcomes with and without the intervention. This involves estimating a causal effect, mirroring the methodology employed in RCTs [17]. However, RCTs encounter challenges such as selection biases

[19,20], difficulties in recruiting diverse populations, and limited sample sizes for exploring treatment heterogeneity across subgroups. Routinely collected data presents a unique opportunity to assess real-life benefit-risk trade-offs associated with a decision [21], with reduced sampling bias and sufficient data to capture heterogeneity [22]. Nevertheless, estimating causal effects from such data is challenging due to the confounding of the intervention by indication. Therefore, dedicated statistical techniques are imperative to emulate a "target trial" [23] from observational data.

## Multiple perspectives on evidence-based decision making

Across different fields, existing literature has emphasized different challenges associated with estimating treatment effects using observational data. While epidemiologic studies underscore the importance of the target trial approach [24–28], there emphasis primarily lies on biases that arise from temporal effects [23,29–33] or confounding variables [34–36], with relatively less attention to issues arising from estimator selection. Recent replications of RCTs using observational data did not explore the impact of modern machine learning methods on the robustness of the results [27,37].

In contrast, machine learning and causal inference literature predominantly studies estimators [38–42] : propensity score matching [43], inverse probability weighting [44], outcome models [45], doubly robust methods, [39] or deep learning based models [46]. This literature may be opaque for some due to intricate mathematical details and unverifiable assumptions. Guidelines seldom address time-related biases, or covariate aggregation which frequently emerge in datasets with temporal dependencies [29,31]. Recently, the machine learning community shifted its focus from EHR data to simulated data, which may not capture the complexities of real-world data [47–50].

In this work, we bring together epidemiological concepts and principles from statistical and machine learning literature. We adopt an empirical perspective to answer practical needs of applied researchers. A study of choices spread out across the analysis –study design, consideration of confounders, and selection of estimators (refer to Section Step-by-step framework for robust decision-making from EHR data)– highlights their equal importance in ensuring the validity of results. To illustrate and compare biases, we investigate the impact of albumin on sepsis mortality using data from a publicly available intensive care database, MIMIC-IV [51] (section Application: evidence from MIMIC-IV on which resuscitation fluid to use).

The primary focus of the main section is on accessibility, with technical details expanded in the appendices.

## Step-by-step framework for robust decision-making from EHR data

Whether or not using machine learning, many pitfalls threaten an analysis' value for decision-making. To avoid these pitfalls, we outline a simple step-by-step analytic framework illustrated in Fig 1 for retrospective case-control studies. We frame the medical question as a target trial [52] to match the design to an RCT giving the gold standard average effect. Then we probe for heterogeneity –predictions on sub-groups– going beyond what RCTs can achieve.

### Step 1: study design – Frame the question to avoid biases

Grounding decisions on evidence needs well-framed questions, defined by their PICO(T) components. Population, Intervention, Control, and Outcome [53,54], and in case of EHRs or

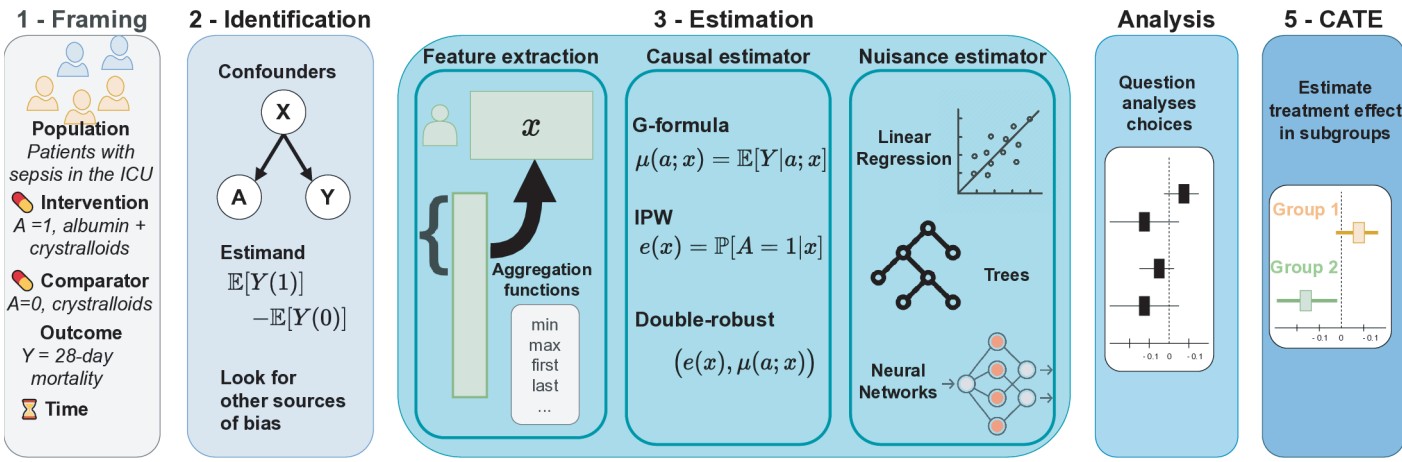

**Fig 1. Step-by-step analytic framework.** The complete inference pipeline confronts the analyst with many choices, some guided by domain knowledge, others by data insights. Making those choices explicit is necessary to ensure robustness and reproducibility.

claims data an additional time component, are necessary to concord with a (hypothetical) target randomized clinical trial [37,55] – Table 1. A selection flowchart such as in S5 Fig makes inclusion and exclusion choices for PICOT explicit.

Without care in defining these PICO(T) components, non-causal associations between treatment and outcomes can easily be introduced into an analysis [56]. The time-varying nature of EHR calls for checking systematically of the Population and Time components by addressing two commonly encountered types of bias.

**Selection Bias.** In EHRs, outcomes and treatments are often not directly available and need to be inferred from indirect events. These signals could be missing not-at random, sometimes correlated with the treatment allocation [57]. For example, billing codes can be strongly associated with case-severity and cost. Consider comparing the effectiveness of fluid resuscitation with albumin to crystalloids. As albumin is more costly, this treatment is more likely to have a sepsis billing code. On the contrary, for patients treated with crystalloids, only the most severe cases will have a billing code. Naively comparing patients would overestimate the effect of albumin.

**Table 1. PICO(T) components help to clearly define the medical question of interest.**

| PICO component | Description | Notation | Example |
|---|---|---|---|
| Population | What is the target population of interest? | $X \sim \mathbb{P}(X)$, the covariate distribution | Patients with sepsis in the ICU |
| Intervention | What is the treatment? | $A \sim \mathbb{P}(A = 1) = p_A$, the probability to be treated | Combination of crystalloids and albumin |
| Control | What is the clinically relevant comparator? | $1 - A \sim 1 - p_A$ | Crystalloids only |
| Outcome | What are the outcomes to compare? | $Y(1), Y(0) \sim \mathbb{P}(Y(1), Y(0))$, the potential outcomes distribution | 28-day mortality |
| Time | Is the start of follow-up aligned with intervention assignment? | N/A | Intervention within the first day |

**Immortal time bias.** Improper alignment of the inclusion defining event and the intervention time is a major source of bias in time-varying data [23,29,32]. Immortal time bias (illustrated in S2 Fig) occurs when the follow-up period, i.e. cohort entry, starts before the intervention, e.g. prescription for a second-line treatment. In this case, the treated group will be biased towards patients still alive at the time of assignment and thus overestimating the effect size. Other frequent temporal biases are lead time bias [30,31] or right censorship [23], and attrition bias [33]. Good practices include explicitly stating the cohort inclusion event [58,Chapter 10:Defining Cohorts] and defining an appropriate grace period between starting time and the intervention assignment [23]. At this step, a population timeline can help.

## Step 2: identification – List necessary information to answer the causal question

The identification step builds a causal model to answer the research question. Indeed, the analysis must compensate for differences between treated and non-treated that are not due to the intervention ([59,chapter 1], [26,chapter 1]).

**Causal assumptions.** Valid causal inference requires assumptions [60] –detailed in S1 Appendix. The analyst should thus review the plausibility of the following: 1) Unconfoundedness: after adjusting for the confounders as ascertained by domain expert insight, treatment allocation should be random; 2) Overlap –also called positivity– the distribution of confounding variables overlaps between the treated and controls –this is the only assumption testable from data [44]–; 3) No interference between units and consistency in the treatment, a reasonable assumption in most clinical questions.

**Categorizing covariates.** Potential predictors –covariates– should be categorized depending on their causal relations with the intervention and the outcome (illustrated in S4 Fig): *confounders* are common causes of the intervention and the outcome; *colliders* are caused by both the intervention and the outcome; *instrumental variables* are a cause of the intervention but not the outcome, *mediators* are caused by the intervention and is a cause of the outcome. Finally, *effect modifiers* interact with the treatment, and thus modulate the treatment effect in subpopulations [61].

To capture a valid causal effect, the analysis should only include confounders and possible treatment-effect modifiers to study the resulting heterogeneity. Regressing the outcome on instrumental and post-treatment variables (colliders and mediators) will lead to biased causal estimates [35]. Drawing causal Directed Acyclic Graphs (DAGs) [34], *eg* with a webtool such as DAGitty [62], helps capturing the relevant variables and defining a suitable estimand or effect measure.

Unconfoundedness –inclusion of all confounders in the analysis– is a strong assumption that can be difficult to ascertain in practice applications. In these cases, sensitivity analyses for omitted variable bias allow to test the robustness of the results to missing confounders [63], proximal inference can be used to leverage proxy of unobserved confounders [64], and the presence of a natural experiment or RCT might identify the desired causal effect without unconfoundedness [65,Chapter 5, 9].

The *estimand* is the final causal quantity estimated from the data. Depending on the question, different estimands are better suited to contrast the two potential outcomes $E[Y(1)]$ and $E[Y(0)]$ [66,67]. For continuous outcomes, risk difference is a natural estimand, while for binary outcomes (e.g. events) the choice of estimand depends on the scale. Whereas the risk difference is very informative at the population level, e.g. for medico-economic decision-making, the risk ratio and the hazard ratio are more informative at the level of sub-groups or individuals [67].

**Causal estimators.**  A given estimand can be estimated through different methods. One can model the outcome with regression models also known as G-formula, [45] and use it as a predictive counterfactual model for all possible treatments for a given patient. Alternatively, one can model the propensity of being treated for use in matching or Inverse Propensity Weighting (IPW) [44]. Finally, doubly robust methods model both the outcome and the treatment, benefiting from the convergence of both models [65]. There is a variety of doubly robust models, reviewed in S2 Appendix.

## Step 3: Statistical estimation – Compute the causal effect of interest

**Confounder aggregation.**  Confounders captured via measures collected over multiple time points must be aggregated at the patient level. Simple forms of aggregation include taking the first or last value before a time point, or an aggregate such as mean or median over time. More elaborate choices may rely on hourly aggregations providing more detailed information on the disease course such as vital signs. They may reduce confounding bias between rapidly deteriorating and stable patients but also increase the number of confounders making estimation more challenging [68]. The increase of variance occurs either in arbitrarily small propensity scores for treatment models or in hazardous extrapolation from one group to another for outcome model. If multiple choices appear reasonable, one should compare them in a vibration analysis (see Step 4: Vibration analysis – Assess the robustness of the hypotheses).

Beyond tabular data, unstructured clinical text may capture confounding or prognostic information [69,70] which can be added in the causal model [28]. However, high-dimensional confounder space such as text may break the positivity assumption just as hourly aggregation choices for measurements.

Missing covariate values might also be a source of confounding. Some statistical estimators (such as forests) can directly incorporate them as supplementary covariates. Others, such as linear models, require imputations. S3 Appendix details general sanity checks for imputation strategies when using statistical estimators.

**Statistical estimation models of outcome and treatment.**  The causal estimators use models of the outcome or the treatment –called nuisances. There is currently no clear best practice to choose the corresponding statistical model [48,71]. The trade-off lies between simple models risking misspecification of the nuisance parameters versus flexible models risking to overfit the data at small sample sizes. Stacking models of different complexity as in a super-learner is a good solution to navigate the trade-off [72,73].

## Step 4: Vibration analysis – Assess the robustness of the hypotheses

Some choices in the pipeline may not be clear cut. Several options should then be explored, to derive conceptual error bars going beyond a single statistical model. When quantifying the bias from unobserved confounders, this process is sometimes called sensitivity analysis [74–76]. Following [77], we use the term vibration analysis to describe the sensitivity of the results to all analytic choices.

## Step 5: Treatment heterogeneity – Compute treatment effects on subpopulations

Once the causal design and corresponding estimators are established, they can be used to explore the variation of treatment effects among subgroups. A causally-grounded model can

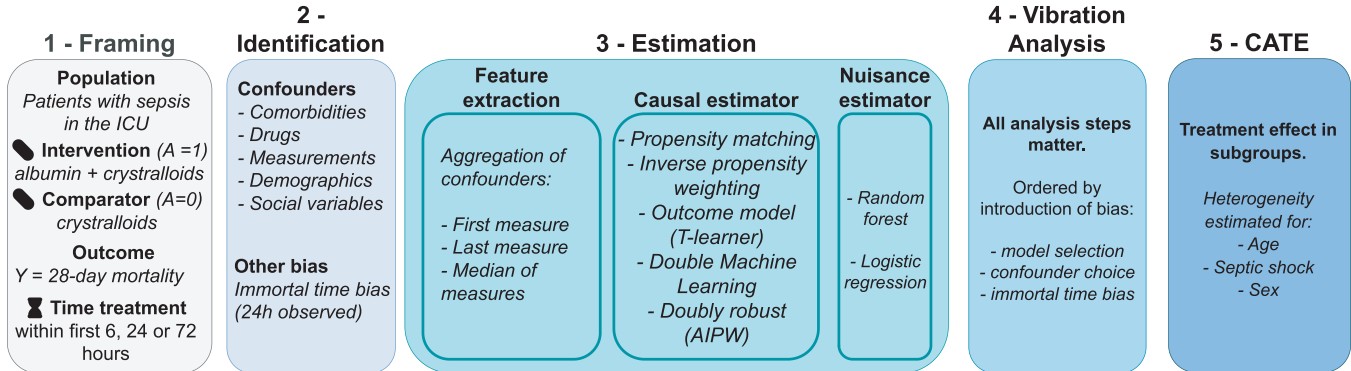

**Fig 2. Application of the step-by-step framework on which resuscitation fluid to use.**

be used to predict the effect of the treatment from all the covariates –confounders and effect modifiers– the *Conditional Average Treatment Effect* (CATE) [78]. Practically, CATEs can be estimated by regressing an individual's predictions given by the causal estimator against the sources of heterogeneity (details in S7 Appendix).

## Application: evidence from MIMIC-IV on which resuscitation fluid to use

We now use the above framework to extract evidence-based decision rules for resuscitation. Ensuring optimal organ perfusion in patients with septic shock requires resuscitation by reestablishing circulatory volume with intravenous fluids. While crystalloids are readily available, inexpensive and safe, a large fraction of the administered volume is not retained in the vasculature. Colloids offer the theoretical benefit of retaining more volume, but might be more costly and have adverse effects [79]. Meta-analyses from multiple pivotal RCTs found no effect of adding albumin to crystalloids [80,81] on 28-day and 90-day mortality. Given this previous evidence, we thus expect no average effect of albumin on mortality in sepsis patients. However, studies –RCT [82] and observational [83]– have found that septic-shock patients do benefit from albumin.

**Emulated trial: Effect of albumin in combination with crystalloids compared to crystalloids alone on 28-day mortality in patients with sepsis.** Multiple published RCTs can validate the analysis pipeline before investigating sub-population effects for individualized decisions. Using MIMIC-IV [51], we compare the magnitude of biases introduced by reasonable choices in the different analytical steps recalled in Fig 2.

MIMIC-IV is a publicly available database that contains information from real ICU stays of patients admitted to one tertiary academic medical center, Beth Israel Deaconess Medical Center (BIDMC), in Boston, United States between 2008 and 2019. The data in MIMIC-IV has been previously de-identified, and the institutional review boards of the Massachusetts Institute of Technology (No. 0403000206) and BIDMC (2001-P-001699/14) both approved the use of the database for research. The database contains comprehensive information from ICU stays including vital signs, laboratory measurements, medications, and mortality data up to one year after discharge.

**Table 2. Characteristics of the trial population measured on the first 24 hours of ICU stay.**

| | Missing | Overall | Cristalloids only | Cristalloids + Albumin | P-Value |
|---|---|---|---|---|---|
| n | | 18421 | 14862 | 3559 | |
| Female, n (%) | | 7653 (41.5) | 6322 (42.5) | 1331 (37.4) | |
| White, n (%) | | 12366 (67.1) | 9808 (66.0) | 2558 (71.9) | |
| Emergency admission, n (%) | | 9605 (52.1) | 8512 (57.3) | 1093 (30.7) | |
| admission_age, mean (SD) | 0 | 66.3 (16.2) | 66.1 (16.8) | 67.3 (13.1) | <0.001 |
| SOFA, mean (SD) | 0 | 6.0 (3.5) | 5.7 (3.4) | 6.9 (3.6) | <0.001 |
| lactate, mean (SD) | 4616 | 3.0 (2.5) | 2.8 (2.4) | 3.7 (2.6) | <0.001 |

S1 Table describes all confounders used in the analysis.

## Step 1: Study design – effect of crystalloids on mortality in sepsis

- Population: Patients with sepsis in an ICU stay according to the sepsis-3 definition. Other inclusion criteria: sufficient follow-up of at least 24 hours, and age over 18 years. S5 Fig details the selection flowchart and S1 Table the population characteristics.
- Intervention: Treatment with a combination of crystalloids and albumin during the first 24 hours of an ICU stay.
- Control: Treatment with crystalloids only in the first 24 hours of an ICU stay.
- Outcome: 28-day mortality.
- Time: Follow-up begins after the first administration of crystalloids. Thus, we potentially introduce a small immortal time bias by allowing a time gap between follow-up and the start of the albumin treatment –see the full timeline in S3 Fig. Because we are only considering the first 24 hours of an ICU stay, we hypothesize that this gap is insufficient to affect our results. We test this hypothesis in the vibration analysis step.

In MIMIC-IV, these inclusion criteria yield 18,121 patients of which 3,559 were treated with a combination of crystalloids and albumin. While glycopeptide antibiotic therapy was similar between both groups (51.8% crystalloid vs 51.5% crystalloids + albumin), aminoglycosides, carbapenems, and beta-lactams were more frequent in the crystalloid only group (2.0% vs. 0.7%, 4.3% vs. 2.6%, and 35.5% vs. 13.8%, respectively). The crystalloid only group was more frequently admitted as an emergency (57.3% vs. 30.7%). Vasopressors (80.2% vs 41.7%) and ventilation (96.8% vs 87.0%) were more prevalent in the treated populations, underlying the overall higher severity of patients receiving albumin (mean SOFA at admission 6.9 vs. 5.7). Table 2 details patient characteristics.

## Step 2: Identification – listing confounders

For confounders selection we use a causal DAG shown in Fig S6 Fig. Gray confounders are not controlled for since they are not available in the data. However, resulting confounding biases are captured by proxies such as comorbidity scores (SOFA or SAPS II) or other variables (eg. race, gender, age, weight). S1 Table details confounders summary statistics for treated and controls.

**Causal estimators.** We implemented multiple estimation strategies, including Inverse Propensity Weighting (IPW), outcome modeling (G-formula) with T-Learner, Augmented Inverse Propensity Weighting (AIPW) and Double Machine Learning (DML). We used the python packages dowhy [41] for IPW implementation and EconML [84] for all other estimation strategies. Confidence intervals were estimated by bootstrap (50 repetitions). S2

Appendix and S4 Appendix detail the estimators and the available Python implementations. S3 Appendix details statistical considerations that we identified as important but missing in these packages, namely lack of cross fitting estimators, bad practices for imputation, or lack of closed form confidence intervals.

## Step 3: Statistical estimation

**Confounder aggregation.**   We tested multiple aggregations such as the last value before the start of the follow-up period, the first observed value, and both the first and last values as separated features. Missing values were median imputed for numerical features, categorical variables were one-hot encoded (thus discarding missing values).

**Outcome and treatment estimators.**   To model the outcome and treatment, we used two common but different estimators: random forests and ridge logistic regression implemented with scikit-learn [85]. We chose the hyperparameters with a random search procedure (S5 Appendix). While logistic regression handles predictors in a linear fashion, random forests bring the benefit of modeling non-linear relations.

## Step 4: Vibration analysis – Comparing sources of systematic errors

**Study design flaw – Illustration of immortal time bias.**   To illustrate the risk of immortal-time bias, we vary the eligibility period of treatment or control in a shorter or longer time window than 24 hours. As explained in Step 1: study design – Frame the question to avoid biases, a longer eligibility period means that patients are more likely to be treated if they survived up to the intervention and hence the study is biased to overestimate the beneficial effect of the intervention. Fig 3a) shows that longer eligibility periods lead to albumin being markedly more efficient (detailed results with causal forest and other choices of aggregation in S8 Fig).

**Confounder choice flaw.**   We consider other choice of confounding variables (detailed in S6 Appendix). Fig 3b) shows that a less thorough choice, neglecting the administrated drugs, makes little to no difference. Major errors, such as omitting the biological measurements or using only socio-demographical variables, lead to sizeable bias. This is consistent with the literature highlighting the importance of a clinically valid DAG [34].

**Estimation choices flaw – Confounder aggregation, causal and nuisance estimators.**   Fig 3c) shows varying confidence intervals (CI) depending on the method. Doubly-robust methods provide the narrowest CIs, whereas the outcome-regression methods have the largest CI. The estimates of the forest models are closer to the consensus across prior studies (no effect) than the logistic regression indicating a better fit of non-linear relationships. We only report the first and last pre-treatment feature aggregation strategy, since detailed analysis showed little differences for other aggregations (S7 Fig for complete results, and S9 Fig for a detailed study on aggregation choices). Both methodological studies [86] and consistency with published RCTs suggest to prefer doubly-robust approaches.

## Step 5: Treatment heterogeneity – Which treatment for a sub-population?

With adequate choice of study design, confounding variables and causal estimator, the average treatment effect matches well published findings: Pooling evidence from high-quality RCTs, no effect of albumin in severe sepsis was demonstrated for both 28-day mortality (odds ratio (OR) 0.93, 95% CI 0.80-1.08) and 90-day mortality (OR 0.88, 95% CI 0.761.01) [80]. Having validated the analytical pipeline, we can use it to inform decision-making. We explore

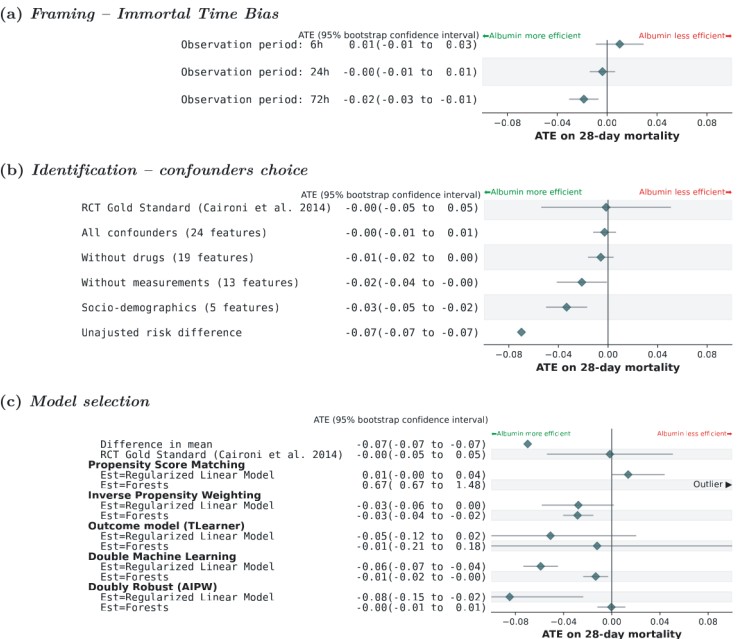

**Fig 3. Framing – Immortal Time Bias**

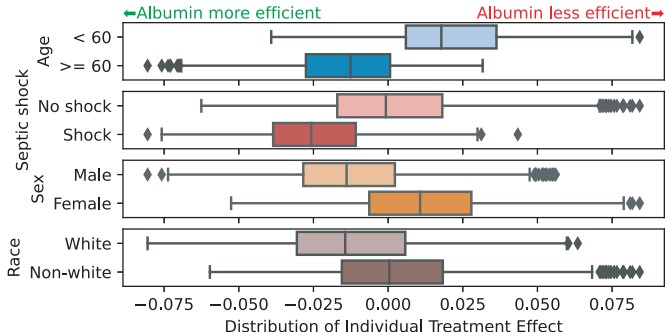

**Fig 4. Subgroup distributions of Individual Treatment effects**: better treatment efficacy for patients older than 60 years, septic shock, and to a lower extent males. The final estimator is ridge regression. The boxes contain the 25th and 75th percentiles of the CATE distributions with the median indicated by the vertical line. The whiskers extend to 1.5 times the inter-quartile range of the distribution.

heterogeneity along four binary patient characteristics, displayed in Fig 4. We find that albumin is beneficial with patient with septic shock consistent with one RCT [82]. It is also beneficial for older patients (age >=60) and males. S7 Appendix details the heterogeneity analysis.

## Discussion and conclusion

Valid decision-making evidence from EHR data requires a clear causal framework. Indeed, machine-learning algorithms have often extracted non-causal associations between the intervention and the outcome, improper for decision-making [11,13,14]. Machine learning studies in medicine often rely on an implicit causal thinking, via a good understanding of the clinical settings. A clear framework helps making sure nothing falls through the cracks.

We have separated three steps important for causal validity: the choice of study design, confounders, and estimators. Regarding study design, major caveats arise from the time component, where a poor choice of inclusion time easily brings in significant bias. Regarding choice of prediction variables, forgetting some variables that explains both the treatment allocation and the outcome leads to confounding bias, that however remains small when these variables capture weak links. Regarding choice of causal estimators, preferring flexible models such as random forests reduces the bias, in particular for doubly-robust estimators. We have shown that all these three steps are equally important: paying no attention to one of them leads to invalid estimates of treatment effect, yet imperfect but plausible choices lead to small biases of the same order of magnitude for all steps. For instance, despite the emphasis often put on choice of confounders, minor deviations from the expert's causal graph did not introduce substantial bias (3b)), no larger than a too rigid choice of estimator. To assert the validity of the analysis, we argue to relate as much as possible the average effect to a reference target trial, even when the goal is to capture the heterogeneity of the effect to individualize decisions. EHRs complement RCTs: RCTs cannot address all the subpopulations and local practices [19,87]. EHRs often cover many individuals, with the diversity needed to model treatment heterogeneity. The corresponding model can then inform better decision-making [17]: a sub-population analysis (as in Fig 4) can distill rules on which groups of patients should receive a treatment. Beyond a sub-group perspective, patient-specific estimates facilitate a personalized approach to clinical decision-making [88].

Since the early 1980ies, researcher investigated the use of colloid fluids in sepsis resuscitation due to their theoretical advantages. However, evidence has long been conflicting. The debate was sparked anew when new synthetic colloid solutions became available, but were later shown to have renal adverse effects [80]. As even large RCTs left unanswered questions, researchers focused on meta-analyses. Here our analysis is in line with the latest two meta-analyses [80,81], as we found no net benefit for resuscitation with albumin in septic patients overall, but a possible slight benefit for patients with septic shock (see Fig 4). While regular meta-analyses not utilizing patient-level data are restricted in their sensitivity analyses, our approach offers the benefit to investigate further potential effect modifiers such as age, sex, or race.

Even without considering a specific intervention, anchoring machine-learning models on causal mechanisms can make them more robust to distributional shift [89], thus safer and fairer for clinical use [18,90]. Yet it is important to keep in mind that better prediction is not per se a goal in healthcare. Establishing strong predictors might be less important than identifying moderately strong but modifiable risk factors as established in the Framingham cohort [91], or optimizing population-wide cost-effectiveness instead of individual treatment effect.

No sophisticated data-processing tool can safeguard against invalid study design or a major missing confounder, loopholes that can undermine decision-making systems. Our framework helps the investigator ensure causal validity by outlining the important steps and relating average effects to RCTs. Causal grounding of individual predictions should reduce the social disparities that they reinforce [6,92,93], as these are driven by historical decisions and not biological mechanisms. At the population level, it leads to better public health decisions. For instance, going back to cardio-vascular diseases, the stakes are to go beyond risk scores and also account for responder status when prescribing prevention drugs.

## Supporting information

**S1 Fig. Motivating example: Failure of predictive models to predict mortality from pre-treatment variables.**
(PDF)

**S2 Fig. Immortal time bias illustration.**
(PDF)

**S3 Fig. Graphical timeline.**
(PDF)

**S4 Fig. Types of causal variables.**
(PDF)

**S5 Fig. Selection flowchart.**
(PDF)

**S6 Fig. Directed Acyclic Graph.**
(PDF)

**S7 Fig. Complete results for the main analysis.**
(PDF)

**S8 Fig. Complete results for the Immortal time bias.**
(PDF)

**S9 Fig. Vibration analysis for aggregation.**
(PDF)

**S1 Appendix. Assumptions: what is needed for causal inference from observational studies.**
(PDF)

**S2 Appendix. Major causal-inference methods: When to use which estimator?**
(PDF)

**S3 Appendix. Statistical considerations when implementing estimation.**
(PDF)

**S4 Appendix. Packages for causal estimation in the python ecosystem.**
(PDF)

**S5 Appendix. Hyper-parameter search for the nuisance models.**
(PDF)

**S6 Appendix. Deviating from expert ignorability – Impact of smaller confounders sets.**
(PDF)

**S7 Appendix. Details on treatment heterogeneity analysis.**
(PDF)

**S1 Table. Complete description of the confounders for the main analysis.**
(PDF)

## Acknowledgments

We thank all the PhysioNet team for their encouragements and support. In particular: Fredrik Willumsen Haug, João Matos, Luis Nakayama, Sicheng Hao, Alistair Johnson.

## Author contributions

**Conceptualization:** Matthieu Doutreligne.

**Data curation:** Matthieu Doutreligne.

**Formal analysis:** Matthieu Doutreligne.

**Investigation:** Matthieu Doutreligne.

**Methodology:** Matthieu Doutreligne, Tristan Struja.

**Project administration:** Matthieu Doutreligne.

**Software:** Matthieu Doutreligne.

**Supervision:** Gaël Varoquaux.

**Validation:** Matthieu Doutreligne.

**Visualization:** Matthieu Doutreligne.

**Writing – original draft:** Matthieu Doutreligne, Tristan Struja.

**Writing – review & editing:** Matthieu Doutreligne, Tristan Struja, Judith Abecassis, Claire Morgand, Leo Anthony Celi, Gaël Varoquaux.

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
