## [Decision Letter · Decision Letter 0]

16 Jan 2024

PDIG-D-23-00449

Step-by-step causal analysis of EHRs to ground decision-making

PLOS Digital Health

Dear Dr. Doutreligne,

Thank you for submitting your manuscript to PLOS Digital Health. After careful consideration, we feel that it has merit but does not fully meet PLOS Digital Health's publication criteria as it currently stands. Therefore, we invite you to submit a revised version of the manuscript that addresses the points raised during the review process.

Please submit your revised manuscript within 60 days Mar 16 2024 11:59PM. If you will need more time than this to complete your revisions, please reply to this message or contact the journal office at digitalhealth@plos.org. Please include the following items when submitting your revised manuscript:

We look forward to receiving your revised manuscript.

Kind regards,

Akhilanand Chaurasia

Section Editor

PLOS Digital Health

Journal Requirements:

1. We ask that a manuscript source file is provided at Revision. Please upload your manuscript file as a .doc, .docx, .rtf or .tex.

Additional Editor Comments (if provided):

Dear Author,

Based on reviewer's suggestion, you are advised for a major revision of your manuscript and satisfactory rebuttals will be highly appreciated

Reviewers' comments:

Reviewer's Responses to Questions

**Comments to the Author**

1. Does this manuscript meet PLOS Digital Health’s publication criteria? Is the manuscript technically sound, and do the data support the conclusions? The manuscript must describe methodologically and ethically rigorous research with conclusions that are appropriately drawn based on the data presented.

Reviewer #1: Yes

Reviewer #2: Yes

2. Has the statistical analysis been performed appropriately and rigorously?

Reviewer #1: Yes

Reviewer #2: Yes

3. Have the authors made all data underlying the findings in their manuscript fully available (please refer to the Data Availability Statement at the start of the manuscript PDF file)?

Reviewer #1: Yes

Reviewer #2: Yes

4. Is the manuscript presented in an intelligible fashion and written in standard English?

PLOS Digital Health does not copyedit accepted manuscripts, so the language in submitted articles must be clear, correct, and unambiguous. Any typographical or grammatical errors should be corrected at revision, so please note any specific errors here.

Reviewer #1: No

Reviewer #2: Yes

5. Review Comments to the Author

Please use the space provided to explain your answers to the questions above. You may also include additional comments for the author, including concerns about dual publication, research ethics, or publication ethics. (Please upload your review as an attachment if it exceeds 20,000 characters)

Reviewer #1: This is an interesting study and the authors have presented a three-step pitfalls retrospective case-control study. However, I believe the paper has some flaws in some text, and I believe this interesting study has not been investigated to its full potential. I've included some comments below.

1) Abstraction: The abstract should be rewritten as a technical summary, while the author's summary should be written in a less technical style for a broader audience.

2) Introduction: The introduction should be improved to better explain the work's motivation and methodology, and the literature review should be more comprehensive, supplementing the advantages and weaknesses of related research. The first sentence of the introduction is hyperbole and not needed. 

3) Related works: The literature review should provide a comprehensive overview of the current state of the arts in the field, outlining the strengths and weaknesses of related research.

4) Results: The experimental section needs clarification and more details, and the analysis and explanation of the test results need to be expanded.

Reviewer #2: The Abstract and Author summary provided by the authors are the same. It would be helpful if the authors can create a distinct non-technical summary under the Author summary section. Other sections are concisely presented.

6. PLOS authors have the option to publish the peer review history of their article (what does this mean?). If published, this will include your full peer review and any attached files.

**Do you want your identity to be public for this peer review?** For information about this choice, including consent withdrawal, please see our Privacy Policy.

Reviewer #1: Yes: Esmael Ahmed

Reviewer #2: No

---

## [Decision Letter · Decision Letter 1]

28 May 2024

PDIG-D-23-00449R1

Step-by-step causal analysis of EHRs to ground decision-making

PLOS Digital Health

Dear Dr. Doutreligne,

Thank you for submitting your manuscript to PLOS Digital Health. After careful consideration, we feel that it has merit but does not fully meet PLOS Digital Health's publication criteria as it currently stands. Therefore, we invite you to submit a revised version of the manuscript that addresses the points raised during the review process.

Please submit your revised manuscript within 60 days Jul 27 2024 11:59PM. If you will need more time than this to complete your revisions, please reply to this message or contact the journal office at digitalhealth@plos.org. Please include the following items when submitting your revised manuscript:

We look forward to receiving your revised manuscript.

Kind regards,

Akhilanand Chaurasia

Section Editor

PLOS Digital Health

Journal Requirements:

Additional Editor Comments (if provided):

Dear Author,

Kindly provide the satisfactory rebuttal of queries raised by reviewers.

Reviewers' comments:

Reviewer's Responses to Questions

**Comments to the Author**

1. If the authors have adequately addressed your comments raised in a previous round of review and you feel that this manuscript is now acceptable for publication, you may indicate that here to bypass the “Comments to the Author” section, enter your conflict of interest statement in the “Confidential to Editor” section, and submit your "Accept" recommendation.

Reviewer #1: (No Response)

2. Does this manuscript meet PLOS Digital Health’s publication criteria? Is the manuscript technically sound, and do the data support the conclusions? The manuscript must describe methodologically and ethically rigorous research with conclusions that are appropriately drawn based on the data presented.

Reviewer #1: (No Response)

3. Has the statistical analysis been performed appropriately and rigorously?

Reviewer #1: (No Response)

4. Have the authors made all data underlying the findings in their manuscript fully available (please refer to the Data Availability Statement at the start of the manuscript PDF file)?

Reviewer #1: (No Response)

5. Is the manuscript presented in an intelligible fashion and written in standard English?

PLOS Digital Health does not copyedit accepted manuscripts, so the language in submitted articles must be clear, correct, and unambiguous. Any typographical or grammatical errors should be corrected at revision, so please note any specific errors here.

Reviewer #1: (No Response)

6. Review Comments to the Author

Please use the space provided to explain your answers to the questions above. You may also include additional comments for the author, including concerns about dual publication, research ethics, or publication ethics. (Please upload your review as an attachment if it exceeds 20,000 characters)

Reviewer #1: Recommendations 

The manuscript presents a comprehensive framework for robust decision-making based on electronic health record (EHR) data. While the framework is well-structured and addresses important methodological considerations, several aspects need revision to enhance its suitability for publication in reputable journals.

1. Clarity and Organization: The manuscript's structure requires refinement to improve clarity and organization. Each section should flow logically and build upon the previous one. Consider restructuring paragraphs for better coherence and readability.

2. Introduction: The introduction should provide a clear context and motivation for the study, outlining the problem addressed, the research's significance, and its objectives. Including a brief literature review to contextualize the study within existing research would strengthen this section.

3. Methodology: The methodology section needs a more detailed description of data sources, study design, and analytical techniques. Assumptions made during the analysis should be clearly stated and justified to ensure transparency and replicability.

4. Results: Present the results clearly and concisely, using appropriate tables, figures, and descriptive statistics. Provide thorough explanations of the findings and their implications, ensuring they are directly related to the research questions.

5. Discussion: Interpret the results thoroughly and discuss their implications about existing literature. Address any study limitations and suggest future research directions. Emphasize the study's contributions to the field and its potential impact on practice or policy.

6. References: Ensure all references are properly cited and formatted according to the journal's guidelines. Include a comprehensive list of references reflecting relevant literature in the field.

7. Language and Style: Pay close attention to language use, grammar, and writing style, adhering to the journal's formatting and style guidelines. Seek feedback from colleagues or mentors to improve clarity and coherence.

7. PLOS authors have the option to publish the peer review history of their article (what does this mean?). If published, this will include your full peer review and any attached files.

**Do you want your identity to be public for this peer review?** For information about this choice, including consent withdrawal, please see our Privacy Policy.

Reviewer #1: None

---

## [Decision Letter · Decision Letter 2]

24 Oct 2024

PDIG-D-23-00449R2

Step-by-step causal analysis of EHRs to ground decision-making

PLOS Digital Health

Dear Dr. Doutreligne,

Thank you for submitting your manuscript to PLOS Digital Health. This manuscript is interesting and valuable and will add significant insight into the application of causal inference in EHR data. After careful consideration, we feel that it has merit but does not fully meet PLOS Digital Health's publication criteria as it currently stands. Therefore, we invite you to submit a revised version of the manuscript that addresses the points raised during the review process.

Please submit your revised manuscript within 60 days Dec 23 2024 11:59PM. If you will need more time than this to complete your revisions, please reply to this message or contact the journal office at digitalhealth@plos.org. There are a few points to consider.

Abstract Revision: Please revise the Abstract to more closely reflect the style of other manuscripts published in PLOS Digital Health. Start with clarifying the significance of the problem and why it needs to be studied. Subsequently, explain your methods, summarize the most important results, and conclude with what those mean.

Relevance of QRISK score: The text related to the QRISK score appears tangentially related to the current study. Please explain the connection or consider including it in a distinct manuscript.

Methodology: In Methods, please describe the inclusion and exclusion criteria and how you handled missing data.

Discussion: The Discussion section is focused on methodology; please incorporate a paragraph discussing what your results add to the body of literature about using albumin vs crystalloid and how your results may help the clinicians select the most beneficial treatment.

We look forward to receiving your revised manuscript.

Kind regards,

Vesela Kovacheva

Guest Editor

PLOS Digital Health

Vesela Kovacheva

Guest Editor

PLOS Digital Health

Additional Editor Comments (if provided):

Reviewers' comments:

6. Review Comments to the Author

Please use the space provided to explain your answers to the questions above. You may also include additional comments for the author, including concerns about dual publication, research ethics, or publication ethics. (Please upload your review as an attachment if it exceeds 20,000 characters)

Reviewer #6: This paper provides a detailed causal analysis framework to be applied on EHRs. The authors provide a good transparency of work and results, and the paper is clear to read.

- The motivation and previous works are well detailed, though the paper is heavily focused on the introduction.

- A short description of MIMIC is missing.

- The appendix contains extraneous information, I suggest filtering it out to only relevant information.

Reviewer #7: The authors propose a causal inference framework tailored on EHR analysis leveraging publicly available data to support their claims. Authors also provide an overview of the limitations of current ML methdologies and the need for a framework more aligned with expert expectations and truthfull to the underlying generating process.

The manuscript is well written and provides a coincise explanation of its goals. 

Furthermore the manuscript focuses on relevant concepts and novel aspects yet poorly explored in a clinical setting.

there are however two major methodological aspects that needs to be addressed first.

Major Review:

- One of the goals of the manuscript is to derive clearer assessments of treatment efficacy to limit bias exposure to the best of experts knowledge. However, a critical component of a causal framework is a fine definition of a suitable DAG which to the very least should be compatible with the observed EHR data. A senstitvity analisys with respect to the choice of a DAG should be thus strongly encouraged in any causal framework and deserves a proper focus in a manuscript such as the one proposed by the authors.

- Unconfoundedness is a strong a hypothesis that rarely can be enforced in real scenarios. There is however an increasing body of litetarure attempting to solve partial identifiability and cases were causal sufficiency can not be granted. I suggest the authors to further expand their mansucripts to included these concepts.

Minor review:

- There has been an issue in the latex export of the paper, please remove the "Missing charachter" warning between the bibliography and the appendix

- There is seems to be some confusion between causal ad statistical concepts, specifically in line 180. The literature consensus is that causal estimands can be evaluated via statistical estimators through identification.

- I suggest the authors to split the causal inference section from the statistical estimations section for clarity

Best regards

---

## [Editor Report · Decision Letter 3]

10 Dec 2024

Step-by-step causal analysis of EHRs to ground decision-making

PDIG-D-23-00449R3

Dear Mr. Doutreligne,

We are pleased to inform you that your manuscript 'Step-by-step causal analysis of EHRs to ground decision-making' has been provisionally accepted for publication in PLOS Digital Health.

Best regards,

Vesela Kovacheva

Guest Editor

PLOS Digital Health

**Additional Editor Comments (if provided):**

The authors have adequately addressed the concerns of the Reviewers and this manuscript is now suitable for publication.